# PEG/PPG-PDMS-Adamantane-Based Crosslinked Terpolymer Using the ROMP Technique to Prepare a Highly Permeable and CO_2_-Selective Polymer Membrane

**DOI:** 10.3390/polym12081674

**Published:** 2020-07-27

**Authors:** Dongyoung Kim, Iqubal Hossain, Yeonho Kim, Ook Choi, Tae-Hyun Kim

**Affiliations:** 1Organic Material Synthesis Laboratory, Department of Chemistry, Incheon National University, 119 Academy-ro, Yeonsu-gu, Incheon 22012, Korea; bluegod94@nate.com (D.K.); iqubal.chem.ru.08@gmail.com (I.H.); 2Research Institute of Basic Sciences, Incheon National University, 119 Academy-ro, Yeonsu-gu, Incheon 22012, Korea; yeonho@inu.ac.kr (Y.K.); ooksclub@inu.ac.kr (O.C.)

**Keywords:** CO_2_ separation, polymer membrane, ring-opening metathesis polymerization, rubbery polymer, adamantane

## Abstract

In this study, precursor molecules based on PEG/PPG and polydimethylsiloxane (PDMS), both widely used rubbery polymers, were copolymerized with bulky adamantane into copolymer membranes. Ring-opening metathesis polymerization (ROMP) was employed during the polymerization process to create a structure with both ends crosslinked. The precursor molecules and corresponding polymer membranes were characterized using various analytical methods. The polymer membranes were fabricated using different compositions of PDMS and adamantane, to determine how the network structure affected their gas separation performance. PEG/PPG, in which CO_2_ is highly soluble, was copolymerized with PDMS, which has high permeability, and adamantane, which controlled the crosslinking density with a rigid and bulky structure. It was confirmed that the resulting crosslinked polymer membranes exhibited high solubility and diffusivity for CO_2_. Further, their crosslinked structure using ROMP technique made it possible to form good films. The membranes fabricated in the present study exhibited excellent performance, i.e., CO_2_ permeability of up to 514.5 Barrer and CO_2_/N_2_ selectivity of 50.9.

## 1. Introduction

The continued use of fossil fuels worldwide has contributed to steadily increasing greenhouse-gas emissions, despite various climate change agreements. Presently, CO_2_ accounts for the largest proportion of greenhouse-gas emissions, making it the greatest contributor to global warming. Furthermore, when CO_2_ is mixed with natural gas, it reduces the overall efficiency of the fuel and must be selectively separated and removed. For these reasons, many researchers have been working to develop carbon capture utilization and storage (CCUS) technologies with the ability to separate, store, and utilize CO_2_ in various processes, with the goal of reducing the levels of atmospheric CO_2_ [1,2,3].

Among such processes, polymer membrane-based gas separation methods have various advantages. They emit fewer hazardous substances than typical techniques, such as amine scrubbing and cryogenic methods, their energy consumption is relatively low, and the design process can be simplified. Using a polymer membrane with a high affinity for CO_2_ could also be a more eco-friendly and economical approach to reducing CO_2_ emissions. However, permeability and selectivity, which are the two primary gas separation performance parameters, tend to have a trade-off relationship with each other, which makes it challenging to manufacture an ideal membrane that improves both parameters at the same time. This relationship can be evaluated using the Robeson plot, a graph that compares the permeability of various materials for each gas, and their selectivity for each gas pair [4,5]. To develop superior high-performance membranes, it is important to understand the intrinsic nature of the materials being used.

A large amount of recent research has been devoted to developing high-performance gas separation membranes, and much of it has focused on rubbery polymers. Rubbery polymers have a glass transition temperature (*T_g_*) below room temperature, and the active movement of their main chains facilitates high gas diffusivity. However, they are highly diffusive to all kinds of gases; thus, they do not exhibit selective diffusivity (so-called “diffusivity-selectivity”), a property governed by kinetic diameter. Instead, they exhibit solubility-selectivity, which is typically governed by gas condensability [6,7].

Poly(ethylene glycol), or PEG, is a typical rubbery polymer, and contains a continuous and repeated structure of ether groups, facilitating strong interactions with CO_2_ through the quadrupole moment of the ether groups. Therefore, CO_2_ solubility in PEG is known to be excellent. However, PEG is also intrinsically crystalline, and the higher its molecular weight, the larger its loss of permeability. This makes it less likely for this rubbery polymer to achieve high performance as a gas separation membrane [6,7,8,9].

Copolymerizing PEG with PPG (Poly(Propylene Glycol)) has been considered as an alternative to pure PEG since the blend can reduce PEG crystallinity. The methyl groups in the PPG structure can interfere with the packing of ethylene glycol chains, weakening crystallinity. Overall, PPG increases the membrane’s amorphous properties, and this structural advantage improves the gas transportability. However, the methyl groups can also cause poorer interactions with polar gas molecules (such as CO_2_), which has been reported to reduce the selectivity of CO_2_/N_2_. Consequently, it is necessary to use a copolymer with an appropriate PEG/PPG ratio [10,11,12,13].

Polydimethylsiloxane (PDMS) is another typical rubbery polymer. PDMS has a very low glass transition temperature (*T_g_*), of around −120 °C, and thus provides very high permeability. For this reason, it is primarily used as a gutter layer in porous support. In addition to its high permeability, PDMS also has excellent stability against heat, good oxidation properties, and has higher solubility of CO_2_. However, it has poor film-forming ability, making it difficult to form a membrane with PDMS without further modification. In PDMS-based copolymers, permeability is greatly increased when the PDMS content is high, but it has been reported that selectivity is then sharply reduced. Therefore, it is necessary to carefully control the PDMS content [14,15,16].

Various studies have been performed to address the above-mentioned problems while attempting to retain the advantages of PEG and PDMS. The most frequently used approaches are crosslinking or copolymerization with other comonomers that have different properties. Here, crosslinking refers to a process where the main chains of polymers are physically or chemically linked to improve the physical properties of PDMS and simultaneously reduce the crystallinity of PEG. For gas separation, although crosslinking results in some loss of permeability, the network structure has also been observed to increase the degree of gas selectivity in the corresponding crosslinked membranes [17,18,19,20,21,22].

From this perspective, we recently developed a crosslinked copolymer of PEG/PPG and PDMS using the ring opening metathesis polymerization (ROMP) method, and then studied the gas separation properties of the corresponding membranes [23]. We found that the crosslinked membranes not only had a high affinity to CO_2_ owing to PEG, and the PDMS had high permeability, but also exhibited a size-sieving capability not commonly found in rubbery polymers. This resulted from a network structure where both ends are crosslinked. Nevertheless, it was not possible to control the crosslinking density in this study owing to the dinorbornene-functionalized precursor molecules for the ROMP reaction. Therefore, further enhancing the permeability of the polymer membranes from these fully crosslinked structures was very difficult.

We aim to further improve the gas separation performance of these newly developed crosslinked PEG/PPG-PDMS membranes by controlling the crosslinking degree of the corresponding copolymers while maintaining their unique network structure.

To this end, in addition to the two precursor molecules (PEG/PPG and PDMS) used to produce the rubbery polymers, we intend to introduce a new precursor molecule in this work, adamantane, a bulky molecule that has only one reacting site (making crosslinking impossible). We expect that the gas transport properties of the corresponding membranes can be further improved because the adamantane-incorporated precursor unit can control the crosslinking density of the corresponding crosslinked terpolymer. That is, the introduction of adamantane was intended to solve the problem of the high crosslinking degree of the ROMP reaction for our previously developed crosslinked rubbery copolymers based on PEG/PPG and PDMS because the adamantane unit will form a void in the existing network structure, providing a new diffusion route for gas molecules (Figure 1a). Therefore, this polymer synthesis strategy is a new method to control the degree of crosslinking in a crosslinked ROMP polymer.

In this study, we prepared terpolymers based on PEG/PPG-PDMS-adamantane through simultaneous copolymerization and crosslinking (1 in Figure 1b). PEG/PPG, which has a high affinity to CO_2_, was used as the main polymer, and PDMS and adamantane were used as precursors to fabricate a gas separation membrane designed to have high permeability and selectively separate CO_2_. More specifically, we designed the polymer using the following strategy. The interaction between CO_2_ and PEG/PPG was expected to induce high solubility of CO_2_, while the rubbery PDMS would contribute to improving the permeability. In addition, the bulky adamantane unit was expected to lower crosslinking density, owing to its non-connected structure, while overall contributing to enhanced gas permeability. Furthermore, a crosslinking method using ROMP was employed for polymerization to improve the physical properties of the rubbery polymers mentioned above. ROMP has the advantage of easy post-modification after polymer synthesis using various monomers with different functional groups [24,25,26,27,28,29].

The morphology and gas separation properties of the resulting crosslinked terpolymer membranes were investigated.

## 2. Materials and Methods

### 2.1. Materials

*cis*-5-Norbornene-exo-2,3-dicarboxylic anhydride (NB) and ethyl vinyl ether (EVE) were purchased from TCI Co., Ltd. (Tokyo, Japan) and used as obtained. Grubbs Catalyst^®^ 2nd Generation, O,O′-bis(2-aminopropyl) polypropylene glycol-block-polyethylene glycol-block-polypropylene glycol (as also known as Jeffamine^®^ ED-2003, *M_n_* = 1900), bis(3-aminopropyl) terminated polydimethylsiloxane (PDMS, *M_n_* = 2500), 1-adamantylamine, and acetic acid were obtained from Sigma-Aldrich Korea (Yongin, Korea) and used without purification.

#### Synthesis of Norbornene-Functionalized Precursor Mmolecules 3, 4, 5

The general procedure for the synthesis of the norbornene-functionalized precursors is as follows. In an oven-dried two-neck round bottom flask equipped with a reflux condenser, nitrogen inlet, and magnetic stirrer bar, NB (**2**) and the corresponding monomers (PEG/PPG, PDMS, and adamantane) were dissolved in acetic acid. The mixture was vigorously stirred and refluxed for 48 h at 130 ℃ in an atmosphere of nitrogen. The resulting solution was cooled and subsequently purified. The detailed procedure is given in the Appendix A.

### 2.2. Membrane Fabrication

All membranes were prepared in a nitrogen-charged glovebox and prepared by the solution-casting method using the corresponding solution in degassed dichloromethane as a solvent. The desired amount of norbornene-functionalized precursor was dissolved in a solvent with the Grubbs catalyst. For example, for the preparation of the x(PEG/PPG:PDMS:Ad) with the composition of x(10:0.5:0.5), 400 mg of NB-PEG/PPG-NB (3) (1.80 mmol), 25.5 mg of NB-PDMS-NB (4) (9.00 × 10^−3^ mmol), and 2.71 mg of NB-Ad (5) (9.00 × 10^−3^ mmol) were dissolved in 6 mL of degassed dichloromethane in a glass vial. Then, the mixture was stirred until it formed a homogenous solution at room temperature. The Grubbs 2nd generation catalyst (2.5 mg, 2.94 × 10^−3^ mmol) was then dissolved in 1 mL of degassed dichloromethane and dropped into the solution, which was then stirred vigorously for 4 min. Subsequently, the solution was carefully poured into the petri dish (made of polypropylene) and dried overnight. To terminate the reaction, dichloromethane containing ethyl vinyl ether was dropped onto the membrane [30,31]. The film was carefully peeled off and dried in a 40 °C vacuum oven for 2 days. The membrane thickness was controlled between 80 and 95 μm.

### 2.3. Characterization and Gas Separation Measurements 

All of the detailed characterization and gas separation measurement methods used here were taken from our previous report [32] and are described in the Appendix A. 

## 3. Results

### 3.1. Synthesis of Precursor Molecules and Membrane Characterization

To prepare crosslinked membranes based on PEG/PPG-, PDMS-, and adamantane-functionalized terpolymers using the ROMP technique, three precursor molecules having PEG/PPG-, PDMS-, and adamantane-functionalized norbornenes (3, 4, and 5, respectively) (hereinafter referred to as “NB” for norbonene, “NB-PEG/PPG-NB” for 3, “NB-PDMS-NB” for 4, and “NB-Ad” for 5) were first synthesized, as shown in Scheme 1. The PEG/PPG-, PDMS-, and adamantane-functionalized amines were then reacted with the norbornene-anhydride to produce the corresponding norbonene diimide (3, 4, and 5). 

The structure of the obtained precursor molecules was characterized using ^1^H NMR and ATR-FTIR spectroscopy. A proton peak, which corresponded to the double bond of norbornene, was observed at around 6.28 ppm in all precursors (3, 4, and 5). As the imidization proceeded, each starting material’s proton peak was found to slightly shift (Appendix A). In addition, other peaks corresponding to anhydride C=O (carbonyl) in 3, 4, and 5 were found to shift to lower wavenumbers as the anhydride was converted to an imide. It was also found that the characteristic peaks arising from each starting material were maintained. For example, the peaks at 2880 cm^−1^ (C–H stretching), 1464 cm^−1^ (CH_2_ scissoring), and 1100 cm^−1^ (C–O–C stretching) in 3 are ascribed to Jeffamine^®^ ED-2003 [33], the peaks at 1258 cm^−1^ (Si-CH_3_ bending), 1010 cm^−1^ (Si–O–Si stretching), and 788 cm^−1^ (Si–CH_3_ rocking) in 4 are ascribed to PDMS [34], and the peaks at 2896 cm^−1^ (C–H stretching), 1450 cm^−1^ (CH_2_ scissoring), 1400 cm^−1^, and 1000 cm^−1^ (CH_2_ bending) in 5 are ascribed to adamantane [35,36], all suggesting the successful synthesis of the intended precursors.

The three precursors (3, 4, and 5) were then polymerized in the presence of a Grubbs catalyst (2nd generation), and in situ membrane casting was used to produce crosslinked random-type terpolymer membranes with three different compositions of (10:0:1), (10:0.5:0.5), and (10:0.75:0.25) molar ratios for PEG/PPG:PDMS:Ad. These were denoted as x(10:0:1), x(10:0.5:0.5), and x(10:0.75:0.25), respectively (Scheme 1). In the present study, the PEG content (or more specifically, the PEG/PPG ratio) was fixed to the ratio that the PDMS and adamantane were set to 10 mol% relative to PEG. This setup was employed to investigate the changes in the degree of crosslinking within the same structure. It was also based on the composition with the best gas separation performance from our previous experimental results, that is, 10 mol% of PDMS relative to PEG/PPG.

Three uniform membranes with different PDMS and adamantane compositions (x(10:0.75:0.25), x(10:0.5:0.5), and x(10:0:1), where each value represents the molar ratios of PEG/PPG:PDMS:adamantane) were obtained, with thicknesses ranging from 80 to 95 µm (Figure 2).

Membrane fabrication without the catalysts under the same conditions was not possible. Instead, the products simply dissolved and solidified (Appendix A). These results confirmed that, despite the use of precursors with a rubbery nature, the formation of a crosslinked structure through ROMP led to an improvement in the physical properties of the corresponding membranes, while effectively improving their film-forming properties.

The fabricated polymer membranes were found to have a crosslinked structure and thus could not be dissolved in any solvents. Their structure was characterized through comparative spectroscopic analysis using the ATR-FTIR, by comparing the intensities of the characteristic peaks of each precursor molecule (Appendix A and Figure 3), as follows.

The 2960 cm^−1^ peaks in all three membranes, corresponding to the C–H stretching vibration of –CH_3_, were attributed to PDMS, and their intensity tended to decrease with decreasing PDMS content, because the number of methyl groups present therein decreased. The C=O peaks, which corresponded to imides, were observed for all three membranes, both at 1768 and 1700 cm^−1^. Further, the common peaks at 1456 cm^−1^ were attributed to the deformation of the –CH_2_ contained in PEG/PPG [30]. In addition, the broad peaks observed at 1184–972 cm^−1^ were attributed to the large number of C–O–C bonds and Si–O–Si bonds repeated in the PEG/PPG and PDMS structures. The characteristic peaks of PDMS were also observed at 1260 cm^−1^, 1018 cm^−1^, and 800 cm^−1^. Notably, their peak intensity tended to decrease with decreasing PDMS content. In particular, it was observed that the –CH_2_ peak attributable to the PEG/PPG unit was shifted to a lower wavenumber (1464 cm^−1^). It is thought that the secondary interactions between the polymer chains decreased as the PEG/PPG chain transformed into an ordered structure from an originally unordered one owing to ROMP crosslinking [37,38]. This observation provides indirect proof that the membranes were synthesized as intended, with respect to the composition of the added precursor molecules. Since adamantane is made of only C–H alkyl chains in its structure, the peak intensity corresponding to C–H stretching, at 2864 cm^−1^, increased with an increase in adamantane content. However, other peaks overlapped with those arising from the many alkyl groups contained in the polymers and thus could not be clearly distinguished from those attributed to other precursors. Based on these results, it was confirmed that crosslinked polymer membranes were successfully synthesized as intended using three precursors, namely PEG/PPG, PDMS, and NB, through ROMP.

The gel-fraction and density of the crosslinked PEG/PPG:PDMS:adamantane-based terpolymer membranes, x(PEG/PPG:PDMS:Ad), were measured and compared with those measured for the poly(NB-Ad), which was fabricated using only the norbornene-terminated adamantane (NB-Ad) 5 and the same ROMP technique (Table 1).

Further, as the adamantane content increased, the gel-fraction tended to gradually decrease, which indicated a reduction in the degree of crosslinking. These results were attributed to the structure of adamantane, where only one end is substituted with NB, in contrast to PEG/PPG or PDMS, in which both ends terminate with NB, as in structures 3 and 4 (Scheme 1).

In addition to the difference in the number of crosslinking sites, the adamantane unit involved in the polymerization process forms an irregular crosslinking structure in the random-type terpolymer, creating an incomplete network structure in the corresponding crosslinked membranes. As a result, reducing the crosslinking degree is expected to improve gas diffusion and enhance gas permeability.

Further, decreasing the gel fraction was expected to lower the density of the membranes through the formation of more voids in the crosslinked structure, but it tended to increase with a decrease in gel fraction values (Table 1). This is attributed to the reduced content of highly flexible PDMS units, which are known to be low-density [39]. Moreover, it was found that the poly(NB-Ad) had a higher density value than x(PEG/PPG: PDMS:Ad)s, strongly suggesting that the density of our crosslinked membranes is highly dependent on the density of each component used.

Next, wide-angle X-ray scattering (WAXS) was performed on the crosslinked x(PEG/PPG:PDMS:Ad) terpolymer membranes (Figure 4). In all three polymer membranes, broad peaks were observed within the measurement range, which indicates their amorphous nature. In addition, characteristic peaks arising from each precursor molecules (**3**, **4**, and **5**) were also observed, further confirming the successful preparation of the desired terpolymers.

Peaks arising from pure PEG are typically reported to be sharp because of the material’s crystallinity; however, the peak corresponding to PEG/PPG in our terpolymer membranes was very broad and covered a wide range, centered around 20°. This was attributed to the fact that, in the present study, the PEG/PPG effectively hindered the crystallinity of PEG, as expected. In addition, the crosslinking process via ROMP further interrupted the packing of PEG/PPG in our crosslinked x(PEG/PPG:PDMS:Ad) terpolymer membranes.

The PDMS peak was observed at around 8°–16° for x(10:0.75:0.25) and x(10:0.5:0.5) membranes, and its peak intensity was found to gradually decrease as its content decreased [x(10:0.75:0.25) > x(10:0.5:0.5) > x(10:0:1)]. Simultaneously, the 2θ value also slightly shifted, from 11.64° to 12.08°, because the flexibility of those polymers was reduced as the PDMS content decreased (Figure 4).

Meanwhile, for the poly(NB-Ad), a broad XRD peak was observed (Figure 4), while the peak was not clearly distinguishable for the x(PEG/PPG:PDMS:Ad) polymer membranes. This was possibly because the peak corresponding to the adamantane unit was masked (or overlapped) by the peak corresponding to the rubbery PEG, owing to its high content.

### 3.2. Thermal Properties of the x(PEG/PPG:PDMS:Ad) Membranes

The thermal properties of the crosslinked x(PEG/PPG:PDMS:Ad) terpolymer membranes were measured and analyzed using thermogravimetric analysis (TGA) (Figure 5). Further, the degradation behaviors of each precursor (NB-PEG/PPG-NB, NB-PDMS-NB, and NB-Ad) and poly(NB-Ad) were measured under the same conditions, and the results were compared with those measured for the x(PEG/PPG:PDMS:Ad) membranes (Appendix A and Table 1).

Initially, the solvents remaining in all the polymer membranes began to evaporate as the temperature increased up to 100 °C, leading to a loss in weight. Other than the degradation induced by solvent evaporation, the first degradation commenced at a relatively high temperature, above 250 °C, for all of the x(PEG/PPG:PDMS:Ad) membranes with different PDMS and adamantane compositions. However, a significant portion of the degradation that occurred within this temperature range was confirmed to be due to PEG/PPG, whose relative proportion was much higher.

In contrast, in the terpolymer x(PEG/PPG:PDMS:Ad) membranes, the amount of PDMS and adamantane was much smaller than PEG/PPG, and thus no noticeable changes were observed in the TGA and their derivative curves. Nevertheless, owing to the two-step degradation of NB-PDMS-NB, it was observed that, as the PDMS content in the polymer membrane increased, the *T_d_* value—which is the temperature at which 5% of the initial weight decomposes—gradually decreased, indicating that decomposition occurs quickly at an early temperature (Appendix A). At the same time, more residue remained at the measurement end temperature of 800 °C for x(10:0.75:0.25). Although adamantane is reported to have high thermal stability [35], its thermal stability here was found to be somewhat lower than that of PDMS (Appendix A). It was therefore observed that the amount of residue at 800 °C tended to gradually increase with the increase in PDMS content owing to its high thermal stability (Figure 5a).

To further investigate the change in weight, which decreased with increasing temperature, the first differentiation curves were calculated (Figure 5b). The main component of the prepared x(PEG/PPG:PDMS:Ad) membranes is PEG/PPG, and it was found that the weight loss rate corresponding to its decomposition was large at about 400 °C. As mentioned above, when the PDMS content in the membrane was high, the weight loss rate was greater at a temperature lower than that of the x(10:0:1). Nevertheless, despite being present in all of the synthesized polymer membranes, adamantane was difficult to distinguish in the differential curve, owing to the overlapping of peaks corresponding to the decomposition of other precursors.

Next, the glass transition temperature (*T_g_*) of the crosslinked terpolymer x(PEG/PPG:PDMS:Ad) membranes was examined using differential scanning calorimetry (DSC) within the temperature range below the degradation temperature (from −70 °C to 100 °C). Although PDMS is reported to have a very low *T_g_* (about −120 °C), in all three membranes, the PDMS’s *T_g_* was not found in the temperature range employed in the present study.

Poly(NB-Ad) also has a high *T_g_* (not observed in the present study but reported to be ~271 °C by other studies) [36], and thus no peaks arising from heat exchange were observed in the applied temperature range. 

In contrast, in the x(PEG/PPG:PDMS:Ad) membranes, a *T_g_* peak arising from the PEG/PPG was observed between −54 to −53 °C. Further, the *T_g_* value tended to gradually increase with decreasing PDMS content (that is, with an increase in adamantane content) although the change was not significant because the difference in composition between PDMS and adamantane was not large enough (Figure 6).

The melting temperature (*T_m_*) of all the crosslinked membranes were observed to be around 15 °C, and these *T_m_* values tended to slowly increase with the increase of adamantane content [e.g., from −30.78 °C for x(10:0.75:0.25) to −30.12 °C for x(10:0:1)].

Here, it is worth noting that both the *T_g_* and *T_m_* of our crosslinked terpolymer x(PEG/PPG:PDMS:Ad) membranes were still lower than room temperature, and they therefore retained their rubbery properties even after crosslinking was performed via ROMP. Accordingly, it is expected that high permeability will be ensured and maintained in practice when applied to gas separation applications (to be discussed later).

### 3.3. Morphological Analysis of the x(PEG/PPG:PDMS:Ad) Membranes

The morphologies of the x(PEG/PPG:PDMS:Ad) membranes were observed using field emission scanning electron microscopy (FE-SEM); further, their surfaces were also examined (Figure 7). As can be seen in the images, sphere-shaped phases were dotted over the entire surface for the x(10:0.75:0.25) and x(10:0.5:0.5) membranes, possibly because phase separation occurred between the hydrophilic region, owing to PEG, and the hydrophobic region, owing to the PDMS structure (Figure 7a,b, respectively) [22,40].

To determine which precursor of the three components (PEG/PPG, PDMS, and adamantane), caused the sphere-shaped phases, energy dispersive spectroscopy (EDS) mapping was performed to investigate the distribution of elements and their relative content (atomic %), as shown in Figure 8 and Table 2. The mapping images confirmed that the brighter phase in the SEM images was primarily composed of Si, which implied that these sphere-shaped phases were PDMS and thus hydrophobic. Furthermore, the morphology of these sphere-shaped phases was found to change as the PDMS content changed, as shown in the images below. More specifically, the higher the PDMS content, the larger and more connected the sphere-shaped phases formed (Figure 8d). It should be noted that, in our previous study based on crosslinked membranes formed from PEG/PPG and PDMS, the size of this sphere-shaped phase gradually increased as the content of PDMS decreased [23]. However, the size tended to decrease when PDMS decreased in the current x(PEG/PPG:PDMS:Ad) membranes. These results are attributed to the fact that the random addition of the adamantane unit to the terpolymers during ROMP hindered the formation of uniform PDMS domains.

In contrast, these sphere-shaped, separated phases were not observed in the crosslinked x(10:0:1) membrane without any PDMS. Instead, it exhibited a uniform morphology, as shown in Figure 7c.

Moreover, using EDS mapping, the relative elemental content analyses indicated that the atomic percentage of Si tended to decrease with decreasing PDMS content (Figure 8 and Table 2). The percentage of O was also calculated to decrease. Further, N was observed in the mapping images, but its calculated percentage was negligible because its content was very low.

### 3.4. Gas Separation Properties

To examine the gas separation properties of the crosslinked terpolymer x(PEG/PPG:PDMS:Ad) membranes with respect to the content of PDMS and adamantane, the single gas permeability of each polymer membrane to CO_2_, N_2_, and CH_4_ was measured at 2 atm and 30 °C, the results of which are presented along with the selectivity data in Table 3. In addition, the results were compared with those for our previously developed crosslinked membrane composed of only PEG/PPG and PDMS (Appendix A) [23].

Firstly, a very high permeability to CO_2_ gas was obtained for these crosslinked terpolymer membranes. In fact, for all three x(PEG/PPG:PDMS:Ad) membranes, the permeability to CO_2_ was much higher than that for other gases, i.e., up to 52 times and 17 times higher than that for N_2_ and CH_4_, respectively. Their particularly high permeability to CO_2_ was attributed to PEG/PPG, which is the main component in x(PEG/PPG:PDMS:Ad) membranes and has a high affinity toward CO_2_.

Notably, the order of permeability (P(CO_2_) > P(CH_4_) > P(N_2_)) was governed not by the molecular size of the gas, i.e., CH_4_ (3.8 Å)> N_2_ (3.64 Å) > CO_2_ (3.3 Å), but by the condensability of each gas, i.e., CO_2_ (195 K) > CH_4_ (149 K) > N_2_ (71 K). This tendency was attributed to the use of rubbery polymers with flexible characteristics. 

Furthermore, the main polymer backbone of our x(PEG/PPG:PDMS:Ad) is rubbery polymers with low *T_g_*, a high degree of selectivity to CO_2_/N_2_ was achieved owing to the highly CO_2_-selective PEG/PPG moiety. The degree of selectivity to CO_2_/N_2_ was expected to decrease from x(10:0.75:0.25) to x(10:0:1) because the degree of crosslinking decreased. However, instead, the degree of selectivity continued to increase. Furthermore, in the x(10:0.5:0.5) membrane, despite decreasing PDMS content, the degree of permeability was found to increase, and a loss of permeability was observed in the x(10:0:1) membrane. This trend differs from that in our previous reports of permeability and selectivity for crosslinked membranes made of only PEG/PPG and PDMS [23], where the permeability increased while the selectivity decreased with an increase in PDMS content.

Therefore, to determine which of the three polymers (PEG/PPG, PDMS, and adamantane) that comprise the x(PEG/PPG:PDMS:Ad) membranes determines their high permeability, the diffusivity coefficient and solubility coefficient of the membranes for each gas were measured, and the results were compared with those for our previously reported crosslinked x(PEG/PPG:PDMS)(1:0.10) membrane, which was made with only PEG/PPG and PDMS, but not with the adamantane unit (Table 4 and Appendix A).

It was found that the x(PEG/PPG:PDMS:Ad) membranes followed the typical solution–diffusion mechanism; that is, changes in the composition of PDMS and adamantane, as well as the crosslinking degree, seemed to affect both the diffusivity and solubility coefficient of the corresponding membranes. Since diffusivity is highly dependent on changes in the polymer structure, changes in the crosslinking degree (and hence in the structure) almost definitely caused changes in the value of the diffusivity coefficient of the corresponding polymer membranes. The diffusivity coefficient was found to change markedly as the degree of crosslinking decreased (i.e., as the adamantane content increased).

However, it was found that the diffusion coefficient values continuously decreased with decreases in crosslinking density [x(10:0.75:0.25) > x(10:0.5:0.5) > x(10:0:1)]. It is thought that the loss of diffusivity occurs owing to the tight network structure gradually collapsing owing to a decrease in the crosslinking degree while the content of flexible PDMS decreases (Figure 1a).

Nevertheless, the diffusivity of the crosslinked x(10:0:1) membrane for N_2_ and CH_4_ was higher than that of our previously developed x(PEG/PPG:PDMS)(1:0.10) membrane, which was made of only PEG/PPG and PDMS [23]. Therefore, as was the goal of this study, it is confirmed that diffusivity can be increased by introducing the bulky and rigid adamantane unit, while also controlling the degree of crosslinking. 

On the other hand, it was also found that the size-sieving capability of our crosslinked terpolymer x(PEG/PPG:PDMS:Ad) membranes was reduced in comparison to our previously developed x(PEG/PPG:PDMS)(1:0.10) membrane. This is because the diffusivity of N_2_ and CH_4_ increased owing to a lower crosslinking degree, an effect that is more prominent in larger gases, such as N_2_ and CH_4_, than in CO_2_.

Further, the permeability of x(PEG/PPG:PDMS:Ad) terpolymer membranes was deemed to be affected by the condensability of gas molecules, and hence was found to be dependent on solubility rather than diffusivity. This is because our x(PEG/PPG:PDMS:Ad) terpolymer membranes used rubbery polymers (Table 4).

Surprisingly, irrespective of the decrease in the degree of crosslinking, it was found that the solubility-selectivity increased continuously with increasing adamantane content. It is thought that adamantane units influence the sorption ability by controlling the degree of crosslinking. Therefore, the complex factors of diffusion and dissolution coefficients can explain the continuous increase in single gas selectivity, even when the permeability changes. 

To compare the gas separation performance of the developed polymer membranes, the data points were plotted on a Robeson plot, and the data are compared with that for typical rubbery polymer-based membranes [11,13,21,26,30,31,41,42,43,44,45] (Figure 9). We found that our x(PEG/PPG:PDMS:Ad) membranes had both higher permeability and CO_2_/N_2_ selectivity than most PEG-based membranes, and a hybrid crosslinking membrane of PEG and silane, surpassing the 2008 Robeson upper bound for CO_2_/N_2_ (Figure 9a). Although our x(PEG/PPG:PDMS:Ad) membranes are mainly composed of PEG/PPG, and hence do not have higher permeability than the PDMS-based membranes, a higher selectivity was still observed for our membranes owing to their crosslinked structure.

On the other hand, the separation performance of our crosslinked x(PEG/PPG:PDMS:Ad) membranes for CO_2_/CH_4_ did not exceed the 2008 Robeson upper bound for CO_2_/CH_4_, which could be because our membrane, consisting mainly of rubbery materials, showed a higher solubility for CH_4_ than N_2_.

## 4. Conclusions

In the present study, crosslinked polymer membranes based on three precursor structures, namely PEG/PPG, PDMS, and adamantane, were synthesized using ring-opening metathesis polymerization (ROMP). The membranes were highly selective and exhibited high permeability to CO_2_. To maintain high permeability while minimizing selectivity loss, the combined content of PDMS and adamantane was adjusted to not exceed 10% of the PEG/PPG content. 

The obtained membranes were characterized using spectroscopy, and their physical properties and gas permeation characteristics were examined. Notably, even though the x(PEG/PPG:PDMS:Ad) membranes were mainly made of rubbery polymers, they were found to have a crosslinked structure due to ROMP, and thus exhibited excellent film-forming properties and stable thermal characteristics. Further, it was found that the diffusivity and solubility of the crosslinked x(PEG/PPG:PDMS:Ad) terpolymer membranes varied depending on the PDMS and adamantane content. In particular, decreasing the degree of crosslinking through the addition of bulky adamantane units improved diffusivity, as intended. As a result, the diffusivity of the crosslinked membranes was found to be dependent both on the flexible PDMS content and on the decreased crosslinking degree. It was also observed that changes in the crosslinked network structure can also affect the sorption ability of the corresponding membranes.

All of the crosslinked terpolymer membranes exhibited good separation performance, which surpassed the Robeson plot for CO_2_/N_2_. Notably, the x(10:0.5:0.5) polymer membrane showed excellent gas separation performance [P(CO_2_) = 514.5 Barrer and CO_2_/N_2_ = 50.9].

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
