# Peer review of "PEG/PPG-PDMS-Adamantane-Based Crosslinked Terpolymer Using the ROMP Technique to Prepare a Highly Permeable and CO_2_-Selective Polymer Membrane"

_polymers, 2020, doi:10.3390/polym12081674_

Round 1
Reviewer 1 Report
This is a sound study on a terpolymer for membrane CO2/N2 and CO2/CH4 separations. The manuscript is well-written, and I'd like to support its publication in this journal after some minor changes. Below are some comments:
- Lines 153-155, the author mentioned terminating reaction by adding DCM containing ethyl vinyl ether to “the membrane” after the solution was dried overnight. I am confused how was the reaction terminated in a membrane, mostly in a solid state. Please make it clear.
- Table 2, the sum of atomic % for x(10:0:1) is not 100%
- The authors need to tune down some of the statements, for example, “Further, their crosslinked structure made it possible to achieve a high degree of selectivity to CO2, to an extent that is difficult to achieve in typical rubbery polymer based membranes.” Other rubbery polymers like PEO, PEGDA and PEGMEA can easily achieve high selectivity more than 50.
Author Response
Q1) Lines 153-155, the author mentioned terminating reaction by adding DCM containing ethyl vinyl ether to “the membrane” after the solution was dried overnight. I am confused how was the reaction terminated in a membrane, mostly in a solid state. Please make it clear.
[Answers to comment 1]
- Thank you for the reviewer’s valuable comment.
- In general, the polymers obtained through the ROMP reaction is quenched using ethyl vinyl ether to deactivate the Ru-catalyst, before being precipitated into poor solvent [Please, see references, Macromolecules 2011, 44, 13, 5075–5078; RSC Adv., 2015, 5, 43581; PNAS May 14, 2019 116 (20) 9729-9734].
- Of course, the homogeneous quenching is effective for the termination of the reaction. But in our case, the crosslinked structure (and in-situ membrane casting) is formed through the ROMP reaction. Therefore, the quenching in a heterogeneous method should be applied due to the insoluble nature of the crosslinked membranes.
- In fact, other studies to prepare rubbery polymers using the ROMP technique also applied the same quenching procedure in a heterogeneous method. Please see our reference 35 and 40.
- Upon the reviewer’s comment, we cited these two references as follows:-Revised: “To terminate the reaction, dichloromethane containing ethyl vinyl ether was dropped onto the membrane. [35,40]”
- -Original: To terminate the reaction, dichloromethane containing ethyl vinyl ether was dropped onto the membrane.
Q2) Table 2, the sum of atomic % for x(10:0:1) is not 100%
[Answers to comment 2]
- Thank you for the reviewer’s comment. This is our mistake, and I apologize for this.
- We have now corrected the value of oxygen atomic % for x(10:0:1) properly.
Q3) The authors need to tune down some of the statements, for example, “Further, their crosslinked structure made it possible to achieve a high degree of selectivity to CO2, to an extent that is difficult to achieve in typical rubbery polymer based membranes.” Other rubbery polymers like PEO, PEGDA and PEGMEA can easily achieve high selectivity more than 50.
[Answers to comment 3]
- Thank you for the reviewer’s valuable comments.
- We agree with the reviewer’s comments in that some rubbery polymers indeed showed very high selectivity (over 50). Indeed, our previous studies also showed excellent selectivity up to 59 in the PEG/PPG ROMP crosslinked membrane [Please, see the reference, ACS Appl. Mater. Interfaces 2020, 12, 27286-27299].Therefore, the manuscript has now been revised as follows:.
- i) In the abstract,
- -Original: “Further, their crosslinked structure made it possible to achieve a high degree of selectivity to CO2, to an extent that is difficult to achieve in typical rubbery polymer-based membranes.”-Revised: “Further, their crosslinked structure using ROMP technique made it possible to form good films.
- ”ii) In 3.4 Gas separation properties,
- -Original: “Furthermore, although the main polymer backbone of our x(PEG/PPG:PDMS:Ad) is rubbery polymers with low Tg, a high degree of selectivity to CO2/N2 was achieved owing to the highly CO2-selective PEG/PPG moiety, along with the unique network structure caused by crosslinking.”
- -Revised: “Furthermore, the main polymer backbone of our x(PEG/PPG:PDMS:Ad) is rubbery polymers with low Tg, a high degree of selectivity to CO2/N2 was achieved owing to the highly CO2-selective PEG/PPG moiety.”

Reviewer 2 Report
The manuscript reported a highly permeable and CO2-selective polymer membrane based on PEG/PPG-PDMS-adamantance terpolymer. In this membrane, the PEG/PPG is expected to induce high solubility of CO2, the rubbery PDMS can improve the permeability, while the bulky adamantance is contributing to lower crosslinking density. The resultant membrane exhibited good CO2/N2 separation performance, i.e. CO2 permeability of up to 514.5 Barrer and CO2/N2 selectivity of 50.9. The authors give a comprehensive analysis about membrane technology for Helium recovery and purification. Overall, the research subject is important. The idea is great and experiments carried out are comprehensive. The article is well written and could be publishable in Polymers.
Author Response
[Answers to the general comments]
- Thank you very much for the reviewer’s favourable comments.
